# LLMs can see and hear without any training

**Kumar Ashutosh** [1 2]  **Yossi Gandelsman** [1 3]  **Xinlei Chen** [1]  **Ishan Misra** [1]  **Rohit Girdhar** [1]

## Abstract

We present MILS: Multimodal Iterative LLM Solver, a surprisingly simple, training-free approach, to imbue multimodal capabilities into your favorite LLM. Leveraging their innate ability to perform multi-step reasoning, MILS prompts the LLM to generate candidate outputs, each of which are scored and fed back iteratively, eventually generating a solution to the task. This enables various applications that typically require training specialized models on task-specific data. In particular, we establish a new state-of-the-art on emergent zero-shot image, video and audio captioning. MILS seamlessly applies to media generation as well, discovering prompt rewrites to improve text-to-image generation, and even edit prompts for style transfer! Finally, being a gradient-free optimization approach, MILS can invert multimodal embeddings into text, enabling applications like cross-modal arithmetic. The code to reproduce MILS is available at https://github.com/facebookresearch/MILS.

## 1. Introduction

Test-time reasoning ability of Large Language Models (LLMs) has emerged as a powerful tool for solving challenging tasks. Recently, OpenAI introduced O1 (OpenAI), a model trained using reinforcement learning to leverage test-time compute for progressively better results, especially on complex math and coding tasks. Even without additional training, LLMs have shown impressive improvements by using test time compute in the form Chain-of-Thought (CoT) reasoning, by rolling out an execution plan to respond to a user's query (Wei et al., 2022; Kojima et al., 2022).

In this work, we leverage this innate reasoning ability in LLMs to solve multimodal understanding and generation tasks, without needing any training! Our approach,

MILS: a **M**ultimodal **I**terative **L**LM **S**olver, uses LLMs as a "GENERATOR" to propose candidate solutions to a given task, and an off-the-shelf multimodal model as a "SCORER" to evaluate the quality of each proposal for the said task. The output of the SCORER is fed back into the GENERATOR to give it feedback, and helps produce the next set of candidates that are more likely to solve the task. This iterative process is run until convergence, or a certain number of steps, and produces an output for the task. We find this simple approach is surprisingly powerful and versatile, working across a variety of tasks and modalities. Using different combinations of GENERATORs and SCORERs, MILS is able to tackle tasks including multimodal captioning, generation, editing, and multimodal arithmetic.

Most prior work for such tasks uses specialized models, often trained on data curated for that task. For instance, zero-shot image captioning models are often still trained on paired image-caption data. MILS, on the other hand, does not need any such training, and exhibits *emergent* zero-shot capabilities. For instance, for image captioning, MILS uses a standard Llama (Dubey et al., 2024) LLM as the GENERATOR, along with CLIP (Radford et al., 2021) as the SCORER. Note that while CLIP is trained on image-text data, it is not trained on clean image-caption data that typical captioning models are trained on. Most vision language models that produce captions leverage CLIP only as an initialization, and require post-training on curated captioning data. Hence, while such models may exhibit zero-shot generalization to a *new data distribution* at test time, MILS exhibits *emergent* zero-shot generalization to the *new task* of captioning.

Furthermore, while there do exist some captioning approaches that do not leverage captioning data (Salewski et al., 2023; Tewel et al., 2022; Shaharabany et al., 2023; Zeng et al., 2024), they are limited to a specific modality and more importantly, the specific task. These approaches typically leverage gradients from CLIP to guide the next token prediction, limiting them to captioning. MILS, on the other hand, seamlessly generalizes to new tasks and modalities by simply swapping out the GENERATOR and SCORER modules. For instance, a GENERATOR constructed by simply chaining an LLM to a Text-to-Image (T2I) model, is able to improve over state-of-the-art T2I models by using the LLM as a "prompt rewriter", a capability not afforded by prior work.

---

[1]Meta AI [2]UT Austin [3]UC Berkeley. Correspondence to: Kumar Ashutosh <kumar.ashutosh@utexas.edu>, Rohit Girdhar <rgirdhar@meta.com>.

*Proceedings of the 42$^{nd}$ International Conference on Machine Learning*, Vancouver, Canada. PMLR 267, 2025. Copyright 2025 by the author(s).

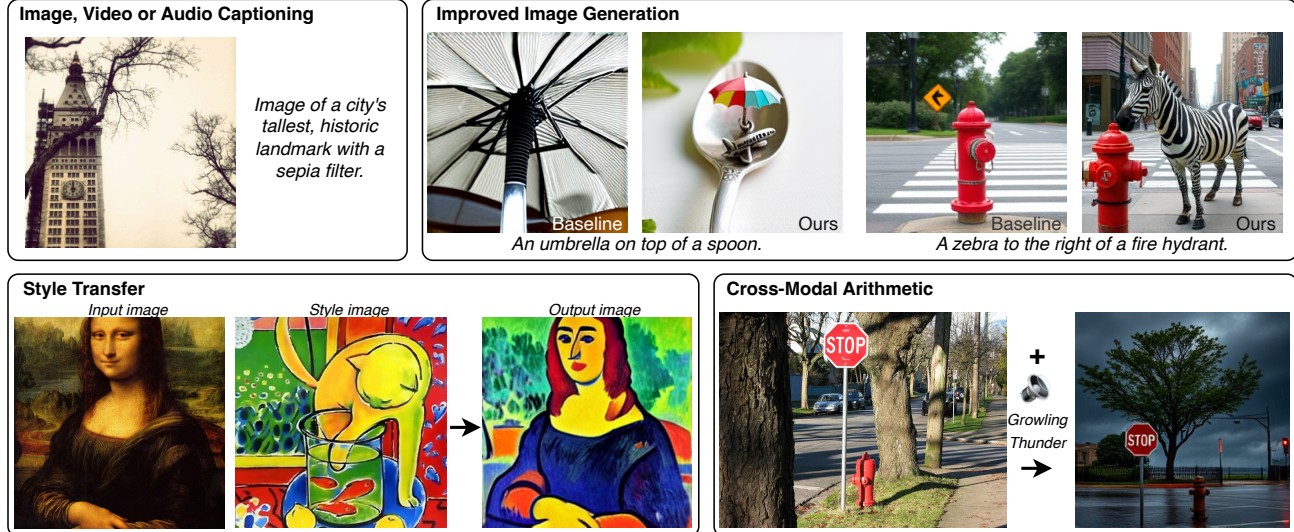

Figure 1: **Our proposed approach, MILS, enables various applications**, from captioning images, video, or audio; improving text-to-image generation; image editing such as style transfer; as well as arithmetic across different modalities by inverting them all into text. It accomplishes all this using a purely test-time optimization approach without any task specific training or data curation!

In this work, we show applicability of MILS across three different visual and non-visual modalities: images, videos and audio, and three different tasks: captioning, generation and editing. Additionally, we show that MILS, being a gradient-free approach, can be used to invert multimodal embeddings into discrete text. This is in contrast to prior work (Kazemi et al., 2024) that uses gradient-based inversion of embeddings into continuous spaces like images. This ability enables novel applications, *e.g.* multimodal arithmetic, by inverting multimodal samples into text, combining them, and mapping them back using MILS's generation ability. We visualize some of these capabilities in Figure 1.

## 2. Related Work

**Multimodal embedding** spaces are typically learned by collecting large amounts of paired multimodal data from the internet, often images and text, and learning encoders for each modality using a pairwise similarity objective (Radford et al., 2021; Ilharco et al., 2021; Zhai et al., 2023; Li et al., 2023a). Such models can further be expanded to additional modalities by collecting text-paired data (Wang et al., 2023; Guzhov et al., 2022), or data paired to a different modality in that embedding space (Girdhar et al., 2023; Gong et al., 2023). These embeddings have enabled various applications, including zero-shot recognition (Radford et al., 2021; Girdhar et al., 2023) open-world object detection (Zhou et al., 2022), and even image generation (Ramesh et al., 2022). We leverage them to compute a similarity score across modalities, which helps guide the optimization, im-

buing multimodal capabilities into an otherwise blind and deaf LLM.

**Generative models** have recently gained popularity due to their ability to generalize to new tasks, often zero-shot. LLMs (Dubey et al., 2024; Jiang et al., 2023; Team et al., 2024) have emerged as the model of choice for discrete input, such as text. Owing to their large size and training on massive corpora, followed by instruction tuning on high quality data often involving human feedback, such models are powerful tools for a variety of tasks. Approaches such as chain-of-thought prompting (Wei et al., 2022; Kojima et al., 2022; Menon et al., 2024), and more recently, training the LLM for better reasoning (OpenAI), have further improved their performance on complex math and coding tasks. However, instruction tuning involves training or fine-tuning LLMs for the target task and modality. In contrast, MILS is an inference-time method that does not require any training and optimizes for the solution at runtime. Recent work has even leveraged LLM's reasoning ability iteratively, to solve optimization (Yang et al., 2023), prompt refinement (Liu et al., 2024), and generation (Mañas et al., 2024) tasks. However, they are either not evaluated on visual tasks (Yang et al., 2023), or shown to have *emergent* zero-shot capabilities (Mañas et al., 2024). Another class of generative models involve diffusion (Nichol & Dhariwal, 2021) or flow-matching (Lipman et al., 2022), popular for continuous domains such as images (Rombach et al., 2022; Betker et al., 2023; Dai et al., 2023; Saharia et al., 2022) and videos (Polyak et al., 2024; Girdhar et al., 2024; Ho et al., 2022; Blattmann et al., 2023). Such models have

dramatically improved in media generation capability, often leveraging LLMs too, for better training data captioning, and inference-time prompt rewrites (Betker et al., 2023; Polyak et al., 2024). Finally, gradient-free approaches (Prasad et al., 2022; Guo et al., 2024; Yang et al., 2023) use the reasoning ability of LLMs to iteratively improve the optimization process.

**Zero-shot multimodal understanding** is studied in two forms: zero-shot across data distributions, and *emergent* zero-shot (Girdhar et al., 2023), where a model generalizes to completely new tasks and not just new data. Multimodal variants of popular LLMs (Dubey et al., 2024; Agrawal et al., 2024; Li et al., 2023b) fall in the former bucket, as they are typically trained or tuned on the kind of data seen at test time. Our focus in this work is the latter, as we show MILS generalizes to completely new tasks at test time. Prior work (Tewel et al., 2022; Zeng et al., 2023; 2024; Salewski et al., 2023; Shaharabany et al., 2023) has attempted this setting, however for specific modalities using specialized techniques. MILS, on the other hand, seamlessly generalizes to many different modalities, across understanding and generation tasks.

## 3. MILS

We now describe our simple approach for solving multimodal tasks using MILS. Since it is training-free, MILS only takes the test sample as input. It relies on two key modules, referred to as the GENERATOR and the SCORER. As the names suggest, GENERATOR generates candidate solutions for the task, while the SCORER scores each of those candidates, and sends them back to the GENERATOR to generate an improved candidate set. For certain tasks, this process may be bootstrapped with scores on an initial candidate set. This optimization is run until convergence or a fixed number of iterations, and produces the final solution to the task. Figure 2 illustrates the overall approach.

**GENERATOR.** The goal of the GENERATOR is to produce candidate outputs $\mathcal{C}$, that solve a given task. It takes as input some text, $\mathcal{T}$, which contains a description of the task, along with scores $\mathcal{S}$ (if any) from the SCORER for the previous optimization step. It leverages this signal to produce the next set of candidate generations. The GENERATOR is typically modeled using an LLM, given its ability to take text as input and reason over it. The output, however, is not limited to text. The candidate generations can be used to prompt a subsequent model to generate other modalities, such as using a text-to-image (T2I) model like Emu (Dai et al., 2023) to generate images. Some GENERATORs could also use the test sample as input, *e.g.* for tasks like image editing or stylization.

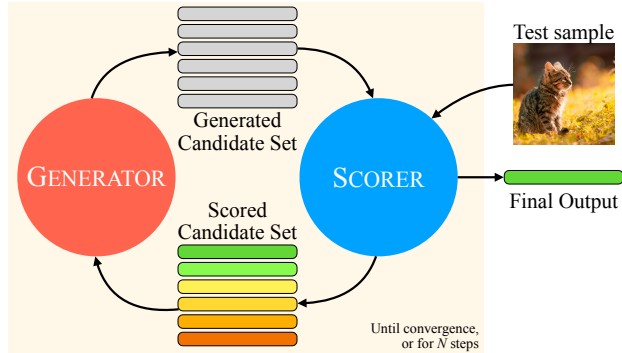

Figure 2: **MILS** leverages two key modules, GENERATOR and SCORER, to solve multimodal tasks. The GENERATOR will generate a number of text candidates, *e.g.* captions for image captioning and prompts for T2I, each of which will be scored by the SCORER, and passed back into the GENERATOR as feedback to generate the next batch of text candidates, eventually producing the final output for the input test sample.

**SCORER.** The goal of the SCORER is to compute a scalar score $\mathcal{S} \in \mathbb{R}$, for the candidates $\mathcal{C}$ from the GENERATOR. It takes as input the test sample, along with $\mathcal{C}$, and compares them. A SCORER could be implemented in various different ways. For instance, it could be a low-level image processing function comparing textures in two images, or it could be a learned machine learning model, such as CLIP (Radford et al., 2021; Ilharco et al., 2021). The SCORER sorts all the candidates based on their score, and returns the top-$K$ candidates along with scores. Depending on the GENERATOR's capacity (context length), the scorer may return the full list of scores, or use an $\epsilon$-greedy strategy to include some low-scoring candidates. In initial experiments, we found greedy top-$K$ to perform the best, hence use that in this work. The output is formatted into the text $\mathcal{T}$, and passed back to the GENERATOR.

**Optimization process.** MILS searches for the optimal generation $\mathcal{C}$ under the SCORER's cost function. The optimization process is run for $N$ steps, or until convergence. Convergence can be defined by the similarity of the candidate set $\mathcal{C}$ over successive steps. Depending on the task, the optimization process can be bootstrapped by an initial candidate set of generations and scoring them. For instance, in case of image captioning, it could simply be a large set of possible image captions from the GENERATOR. For other tasks like T2I, one does not need such an initial set.

## 4. Experiments

We now empirically evaluate MILS and compare it to existing approaches on some of the multimodal understanding and generation tasks enabled by it. For each of the down-

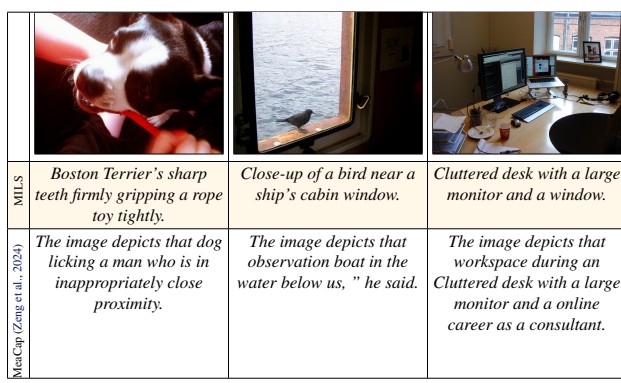

| | | | |
|---|---|---|---|
| MILS | *Boston Terrier's sharp teeth firmly gripping a rope toy tightly.* | *Close-up of a bird near a ship's cabin window.* | *Cluttered desk with a large monitor and a window.* |
| MeaCap (Zeng et al., 2024) | *The image depicts that dog licking a man who is in inappropriately close proximity.* | *The image depicts that observation boat in the water below us, " he said.* | *The image depicts that workspace during an Cluttered desk with a large monitor and a online career as a consultant.* |

Figure 3: **Image Captioning using MILS**, compared to existing state-of-the-art zero-shot approach, MeaCap (Zeng et al., 2024). MILS, while being a much simpler approach, produces more accurate and syntactically correct captions to the image.

| Method | BLEU$_4$ | CIDEr | METEOR | SPICE |
|---|---|---|---|---|
| ZeroCap (Tewel et al., 2022) | 2.6 | 14.6 | 11.5 | 5.5 |
| ConZIC (Zeng et al., 2023) | 1.3 | 13.3 | 11.2 | 5.0 |
| CLIPRe (Li et al., 2023c) | 4.6 | 25.6 | 13.3 | 9.2 |
| MeaCap$_{TF}$ (Zeng et al., 2024) | 7.1 | 42.5 | 16.6 | 11.8 |
| MeaCap$^*_{TF}$ (Zeng et al., 2024) | 4.5 | 26.0 | 14.1 | 9.4 |
| MILS | **8.0** | **33.3** | **15.0** | **9.6** |

Table 1: **Zero-shot image captioning** on MSCOCO (Karpathy & Fei-Fei, 2015). Despite being far simpler than existing approaches, MILS performs competitively on all automatic metrics, and especially METEOR and SPICE which take into account the semantic similarity. *refers to results we obtained by running the provided code.

stream applications, we describe the GENERATOR, SCORER, benchmarks and evaluation setup, followed by the key results. Finally in Section 4.7 we ablate the various design choices in MILS.

Note that MILS is a test-time optimization method and exhibits *emergent zero-shot* behavior (Girdhar et al., 2023), generalizing not only to a new test data distribution, but to the new task and modality itself. This is contrast to most existing zero-shot work that typically needs task/modality-specific data curation or training. Since most prior work is of the latter type, it is hard to perform fair comparisons. Nevertheless, we compare to the closest zero-shot approaches and show that MILS is competitive or better, even compared to methods tuned for that specific task or modality.

### 4.1. Image Captioning

We start with the fundamental image understanding task of producing a textual caption for a given image.

| Method | Training Data | CIDEr | METEOR |
|---|---|---|---|
| Nagrani *et al.* (Nagrani et al., 2022) | HowTo100M (Miech et al., 2019) | 0.5 | 8.23 |
| Nagrani *et al.* (Nagrani et al., 2022) | VideoCC3M (Nagrani et al., 2022) | **8.2** | 11.3 |
| MILS | – | 2.3 | **14.4** |

Table 2: **Zero-shot video captioning** on MSR-VTT (Xu et al., 2016). MILS outperforms (Nagrani et al., 2022) when trained on HowTo100M, and is competitive to it when trained on the much cleaner VideoCC3M dataset, outperforming it on METEOR. We grayed (Nagrani et al., 2022) since it is trained for video captioning, while MILS is not.

**GENERATOR.** We use the Llama 3.1 8B (Dubey et al., 2024) LLM as the core generation module. We generate an initial list of 30K prompts that we use to bootstrap the optimization process. To ensure diversity in this initial set, we prompt the LLM with different object categories to generate a list of prompts, and combine them, similar to (Gandelsman et al., 2024). Then for each optimization step, we keep the top-50 highest scoring generations from the SCORER, and convert them into a textual prompt. The prompt used is described in Section B. We run the optimization process for 10 steps.

**SCORER.** We score the candidate captions against the test image using an image-text similarity model, SigLIP (Zhai et al., 2023). Note that unlike image captioning models that leverage curated image-text pairs (Karpathy & Fei-Fei, 2015), SigLIP, by itself is not capable of captioning (Shen et al., 2022). Nevertheless, combined with MILS, it can serve as an effective captioner, as shown next.

**Benchmarks and Metrics.** We evaluate MILS on the MSCOCO captioning test set (Karpathy & Fei-Fei, 2015). It consists of 5,000 images sampled from the MSCOCO dataset (Lin et al., 2014). We use the standard suite of captioning evaluation metrics, including BLEU (Papineni et al., 2002), METEOR (Banerjee & Lavie, 2005), CIDEr (Vedantam et al., 2015) and SPICE (Anderson et al., 2016). We focus our attention on the METEOR and SPICE metrics, as those take into account semantic similarity rather than exact word match, and are better correlated with human preference (Anderson et al., 2016). This is particularly important for emergent zero-shot approaches like MILS, which are not trained to learn the vocabulary used in a given benchmark or modality.

**Results.** We compare MILS to existing baselines in Table 1. Some of the baselines, such as ZeroCap (Tewel et al., 2022), also leverage language models in conjunction with a CLIP-like model. However, they propose a gradient based optimization process to search for the optimal next token given a current generation. Other approaches like MeaCap (Zeng et al., 2024) filter key concepts from a memory module, and leverage a number of text and multimodal encoders in a multi-step process to produce the caption. In

| Method | BLEU$_4$ | ROUGE$_L$ | METEOR | SPICE |
|---|---|---|---|---|
| ZerAuCap (Salewski et al., 2023) | **2.9** | **25.4** | 9.4 | 5.3 |
| MILS | 2.7 | 23.1 | **12.4** | **7.6** |

Table 3: **Zero-shot audio captioning** on Clotho (Drossos et al., 2020) dataset. MILS performs competitively to the existing zero-shot audio captioning approach ZerAuCap, even outperforming it on semantics-aware metrics like ME-TEOR and SPICE, while being simpler and applicable to many other modalities and tasks.

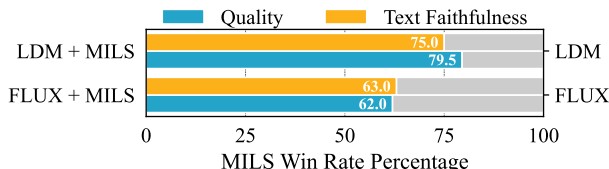

Figure 4: **Improved text-to-image (T2I) generation using MILS.** We apply MILS to two of the latest, state-of-the-art T2I models, a latent diffusion model (LDM), and FLUX.1 [schnell] (FLUX). We compare MILS's outputs to the generations from the initial models using human annotators. Evaluated over the 200 prompt DrawBench dataset, the annotators clearly preferred MILS's generations on both overall quality and text faithfulness, across both models.

contrast, MILS is simpler conceptually and to implement, while obtaining better results. We also show examples of the captions generated using MILS, and compare them to MeaCap in Figure 3. MILS, without ever having seen any captioning data or having captioning-specific training is able to generate faithful and syntactically correct captions. This result also showcases the strong reasoning capabilities of the GENERATOR, which is able to correctly modify the captions for future iterations.

## 4.2. Video Captioning

Owing to its simplicity and versatility, MILS seamlessly transfers to videos without major changes. We use the same GENERATOR as described for image captioning in Section 4.1, along with the same initial prompt set. For SCORER, we use a ViCLIP (Wang et al., 2023) ViT-L/14 model that operates on 8 frames from the video, and returns a similarity score between the video and the caption. We experiment on the MSR-VTT (Xu et al., 2016) test set, which contains 2,990 videos, each between 10 to 30 seconds long. We report our results in Table 2. Since most prior work in video captioning leverages video-caption training data, we compare MILS to (Nagrani et al., 2022), which learns a vision-language model on the HowTo100M (Miech et al., 2019) or the VideoCC3M (Nagrani et al., 2022) datasets, and reports performance on MSR-VTT. We use the CIDEr (Vedantam et al., 2015) and METEOR (Banerjee & Lavie, 2005) metrics as reported in the prior work. We find that MILS, in spite of never having been trained for video captioning, outperforms (Nagrani et al., 2022) trained on HowTo100M on both metrics. Even when compared to the same model trained on the much cleaner VideoCC3M data, MILS outperforms it on the semantics-aware METEOR metric. This difference in performance of the baseline in the two settings shows the importance of training data for video captioning models. MILS, being competitive with these without needing any video captioning training, is very promising. We show qualitative results in the Section C.

## 4.3. Audio Captioning

Similar to videos, MILS transfers seamlessly to audio captioning as well. We use the same GENERATOR as in Section 4.1, along with 50K initial audio prompts generated using an LLM (*cf*. Section B). As the SCORER, we use the ImageBind (Girdhar et al., 2023) model, that maps multiple modalities, including audio and text, to a shared embedding space. We evaluate our approach on a popular audio captioning dataset, Clotho (Drossos et al., 2020). We use automatic captioning metrics as used in prior work, and described in Section 4.1. We report our performance in Table 3. MILS obtains strong performance *vs*. a comparable zero-shot approach, ZerAuCaps (Salewski et al., 2023), outperforming it especially on semantics-aware metrics like METEOR and SPICE. While other approaches for audio captioning have been proposed, they require training on audio-caption data (Kong et al., 2024). See Section C for qualitative results.

## 4.4. High-Quality Image Generation

As mentioned earlier, MILS is not limited to multimodal understanding tasks discussed so far. We now describe how MILS can be used for multimodal generation tasks, starting with improving text-to-image (T2I) generation models.

**GENERATOR.** To generate high quality images, we chain an LLM to a T2I model. Specifically, we experiment with two state-of-the-art models, a Latent Diffusion Model (LDM) (Rombach et al., 2022) and FLUX.1 [schnell] (Labs). The goal of the LLM is to "rewrite" the prompt passed into the T2I model, such that the final generated image improves in quality, while maintaining or improving the faithfulness to the original text prompt. Note that this GENERATOR does not need an initial prompt set to bootstrap from.

**SCORER.** We score the generations using PickScore (Kirstain et al., 2023). PickScore is a

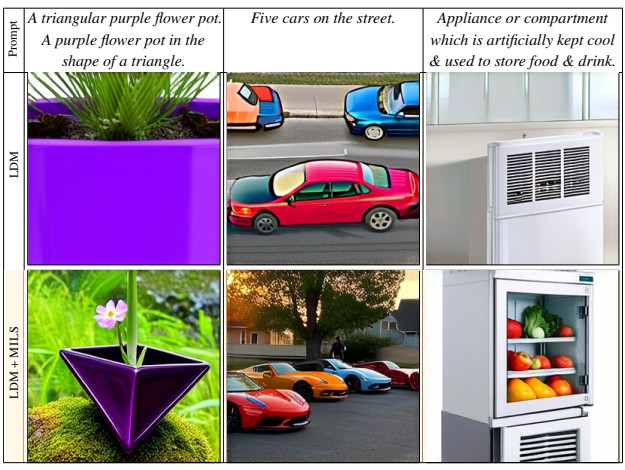

Figure 5: **Improving image generation using MILS.** Applying MILS to a GENERATOR using the same base model, a Latent Diffusion Model (LDM) in this case, leads to much higher quality images. We show the original input prompt, the generation from the base model, and from MILS.

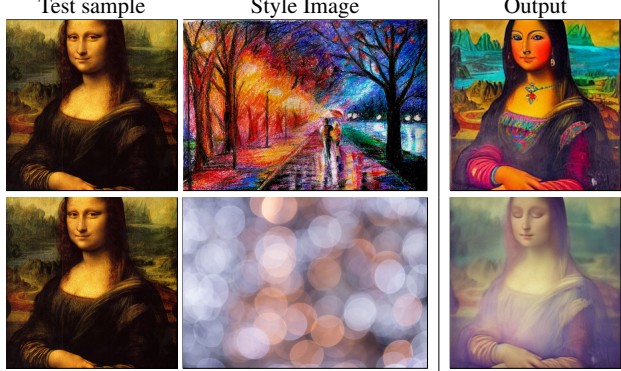

Figure 6: **Style Transfer.** Using Gram Matrix distance (Gatys, 2015) as the SCORER, MILS can discover the edit prompt required to apply a given style to an image.

CLIP-style model takes as input an image and a text prompt, and predicts the likelihood of that image to be preferred by humans for that prompt. We score each of the GENERATOR outputs using PickScore along with the input prompt, and return the scores for each of the generations. Rest of the process proceeds the same as before.

**Benchmarks and Metrics.** We use the DrawBench prompt set from Imagen (Saharia et al., 2022), that contains 200 textual prompts, to evaluate our generations. Due to the noisiness of automatic metrics for media generation (Girdhar et al., 2024; Ge et al., 2024; Jayasumana et al., 2024), we perform our evaluations using human annotators from Amazon Mechanical Turk. We follow the JUICE framework (Girdhar et al., 2024). In-keeping with the standard practice in media generation (Dai et al., 2023; Girdhar et al.,

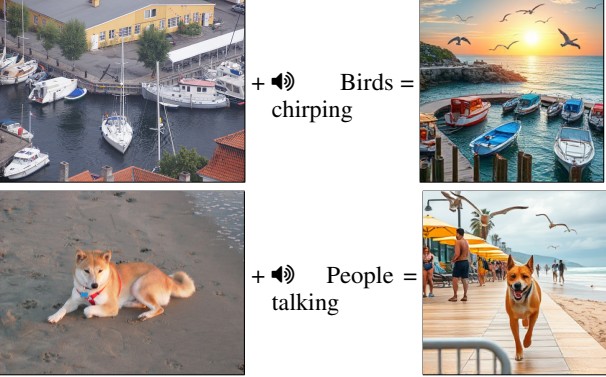

Figure 7: **MILS enables cross-modal arithmetic** by inverting modalities into text, combining them, and mapping them back to an image.

2024), we evaluate the generations on two axes: quality or visual appeal, and the faithfulness to the input text. For each axis, we set up a web interface where an annotator is shown two images, generated either by a baseline model, or enhanced using MILS. When evaluating for text faithfulness, the annotator is also shown the original text prompt. The annotator has to pick which image they prefer. Each image is annotated by three annotators and we use the majority vote over the annotators to compute a win% for each model. See Section A for full human evaluation details.

**Results.** We summarize the results of our human study in Figure 4. As seen by the win-rates, human annotators clearly prefer generations enhanced by MILS over the generations from the base model itself. We also show qualitative comparisons in Figure 5, where the improvement in aesthetic quality using MILS is clearly apparent. We find that MILS is able to simplify complex prompts and add aesthetic details, that improve the overall quality and faithfulness of the generations. Recent work has shown that LLM-based prompt rewrites (Betker et al., 2023; Polyak et al., 2024) can improve media generation performance. However, they require a laborious process of manually trying various different rewrite prompts, until one finds a rewrite that works well with that model, across all test cases. MILS can automate and complement that process, either by prompt-engineering each generation, or proposing candidate rewrites for an expert prompt engineer to improve upon.

Note that this capability is not easily afforded by prior work we compared to on earlier tasks (Tewel et al., 2022; Salewski et al., 2023). Those methods would require computing gradients through a multi-step diffusion process to estimate which tokens the LLM should produce next. MILS, being a gradient-free optimization approach, easily enables such diverse applications within a simple framework.

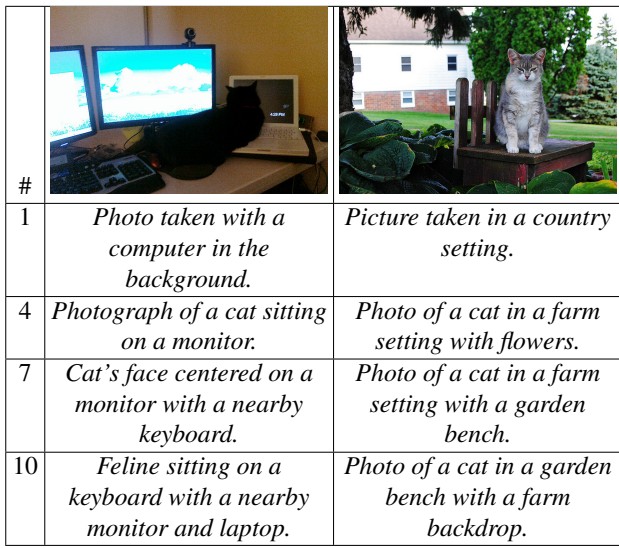

| # | | |
|---|---|---|
| 1 | *Photo taken with a computer in the background.* | *Picture taken in a country setting.* |
| 4 | *Photograph of a cat sitting on a monitor.* | *Photo of a cat in a farm setting with flowers.* |
| 7 | *Cat's face centered on a monitor with a nearby keyboard.* | *Photo of a cat in a farm setting with a garden bench.* |
| 10 | *Feline sitting on a keyboard with a nearby monitor and laptop.* | *Photo of a cat in a garden bench with a farm backdrop.* |

Figure 8: **Image captioning over steps.** This shows the captions generated by MILS at different steps (#) of optimization process, getting progressively more accurate.

### 4.5. Style Transfer

Going beyond image generation, MILS can also be applied to image editing tasks. Here we specifically consider the task of style transfer, where given a test image and a style image, the goal is to generate an image that contains the content from the test image, in the style of the style image.

GENERATOR. Similar to Section 4.4, we implement the GENERATOR by chaining the output of an LLM to an image generation model. Different from Section 4.4, since we want to generate the same content as in the test sample, the GENERATOR also takes the test sample as an input. Hence, we use an image editing model (Sheynin et al., 2024) as the image generation module. It produces the stylized image given the test sample and the edit prompt from the LLM.

SCORER. To measure the quality of the style transfer, we use a simple approach to estimate the similarity of colors and textures in the generated image compared to the style image. We use the distance between Gram matrices of the image features, as proposed in (Gatys, 2015). We compute this distance over features from different layers of a VGG19 (Simonyan & Zisserman, 2015) CNN, where the lower layers ensure stylistic faithfulness, and the higher layers ensure content faithfulness. We use MILS to minimize both the style and content losses.

**Results.** Figure 6 shows some sample style transfer results. MILS generalizes to this novel task completely zero-shot and produces accurately stylized images. Note that it achieves such edits not only without any training, but also without the LLM actually seeing any features from either the test sample, or the style image!

### 4.6. Cross-Modal Arithmetic

Finally, we explore an interesting application enabled by MILS. Unlike prior work (Kazemi et al., 2024) that maps embeddings to continuous image space, our gradient-free approach in MILS enables inverting such embeddings into the discrete text space instead. This is also exemplified by our results in Sections 4.1 to 4.3. This enables an interesting application of cross-modal arithmetic. We take inspiration from ImageBind (Girdhar et al., 2023), which mapped multiple different modalities into the image embedding space. Using this shared embedding, authors were able to combine modalities, and generate or retrieve images given that combination. MILS, in fact, is even more flexible, as inversion to text enables interfacing with many more models. For instance, ImageBind showed results on audio to image generation by leveraging a DALLE-2-like T2I model (Ramesh et al., 2022). This was possible as ImageBind happened to be aligned to the CLIP embedding space, same as what was used in DALLE-2. As such, ImageBind was not compatible with any other T2I model, such as a latent diffusion model (Rombach et al., 2022). A textual representation, on the other hand, would work with any T2I model, including those that do not represent textual input as a point on an embedding space. In Figure 7, we show examples of combining image and audio modalities. We first invert both image and audio into text using Section 4.1 and Section 4.3 respectively, combine the two outputs using an LLM (details in the Section B), and finally convert the prompt into a high quality image as described in Section 4.4. The resulting generated image combines the semantic concepts from both those modalities.

### 4.7. Ablations

We now ablate some key design choices in MILS. We primarily focus on the image captioning task for most of the analysis, and improved image generation for some of the ablations. For computational ease, we randomly sample 1000 images from MSCOCO for captioning, and use the 200 prompt DrawBench set for image generation, as the test set for this analysis. We report all metrics including CLIP similarity and PickScore averaged over these sets.

**Performance over optimization steps.** We evaluate this for both the tasks in Figure 9. We report both the SCORER output, which can be thought of as a "training loss" in our setup, as well as a downstream metric. We report SPICE (Anderson et al., 2016) for image captioning, and human evaluation against the original prompt's generation for T2I. For the latter, we also show ±4 point error bars, which we found to be the typical random variance margins in human evaluations. As Figure 9 shows, both the SCORER outputs and the down-

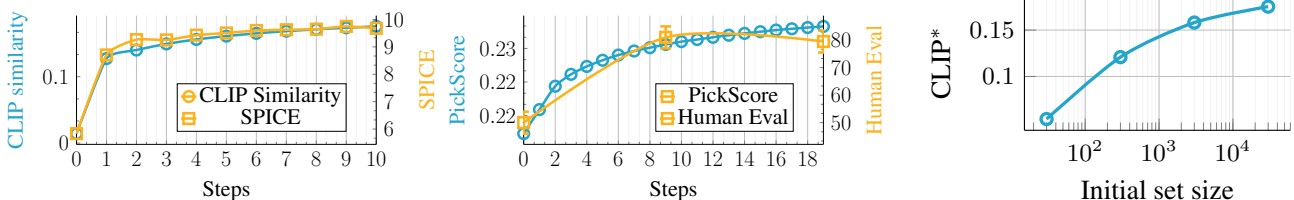

Figure 9: **Performance with number of optimization steps for captioning (left) and generation (right).** We show the optimization metric, *i.e.* SCORER's output (CLIP Similarity and PickScore), and the downstream metric (SPICE (Anderson et al., 2016) and human evaluation win% for quality), respectively. Both tend to improve over optimization steps, and correlate with each other.

Figure 10: **Performance improves with size of initial set**, suggesting it is crucial in bootstrapping the GENERATOR. CLIP*: CLIP similarity at convergence.

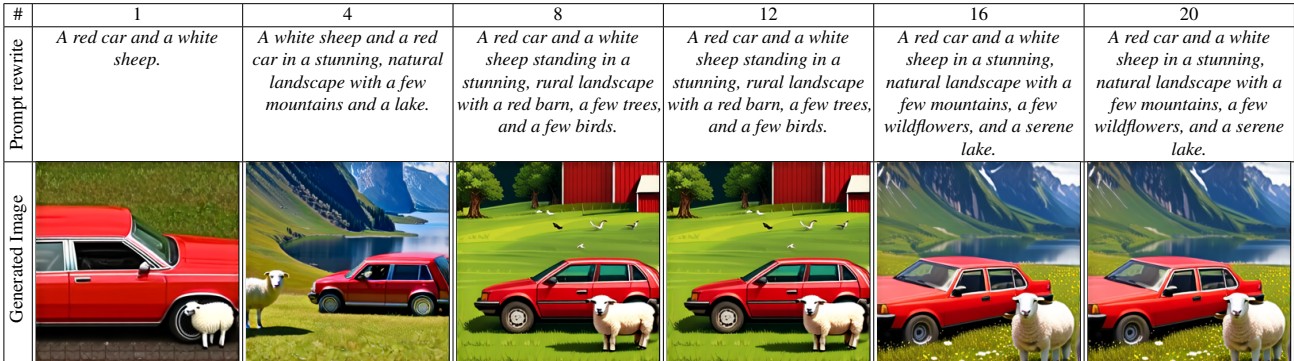

Figure 11: **Improved image generation over optimization steps.** The quality of the output improves over the optimization steps (#). We also show the prompt being produced by the LLM in the GENERATOR, which is passed to the T2I model.

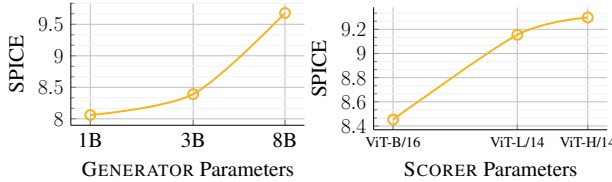

Figure 12: **Impact of GENERATOR (left) and SCORER (right) size.** We report the downstream metric SPICE for both these ablations since CLIP similarity may not be comparable across model sizes. For the GENERATOR and SCORER we use different sized Llama or MetaCLIP models. As evident from the graphs, larger models perform better.

stream metrics improve over optimization steps, converging after 10 to 20 steps. We also note that the optimization objective (SCORER output) correlates well with downstream performance. Lastly, we visualize this result qualitatively in Figures 8 and 11 for captioning and generation respectively. In both cases, the quality of the output improves over the steps, showing the effectiveness of MILS.

**Impact of the initial candidate set.** We evaluate this in Figure 10. We subsample the initial set to different sizes, and see a strong positive correlation between that and the final

performance. This suggests that the initial bootstrapping set is critical to ensure the GENERATOR can produce sufficiently diverse candidates and avoid local minimas.

**Size of the GENERATOR and SCORER.** We evaluate the effect of the size (in parameters) of the GENERATOR (Llama 3) and SCORER (MetaCLIP (Xu et al., 2024)) in Figure 12, for image captioning. We find larger models generally performed better, with LLM scaling exhibiting the most promising gains. We also experiment with different kinds of GENERATORs and SCORERs in the Section C.

## 5. Conclusion and Future Work

We have presented MILS, a simple approach for solving multimodal tasks without requiring any task specific data curation or training. MILS exhitbits emergent zero-shot generalization to various different tasks and modalities. Notably, we show MILS obtains strong performance on captioning across three modalities: images, videos and audio, showing that LLMs can see and hear without any training! This further leads to improving and enabling various media generation tasks, such as image generation, image editing (style transfer) and cross-modal arithmetic.

While promising, MILS has some limitations that future work can attempt to address. Its performance is bounded by the ability of the GENERATOR to generate diverse candidates, and the SCORER to provide accurate feedback to the GENERATOR. For instance, style transfer performance is limited by the resolution of Gram matrix distance in detecting fine-grained texture similarities, and the LLM's ability to describe potential styles. As the LLMs and the multimodal models continue to improve (OpenAI; Fang et al., 2024), MILS would improve with it. Another limitation is the speed of the optimization process. This would improve as the core LLMs become faster and more efficient, and as their context length (Munkhdalai et al., 2024) and reasoning abilities (OpenAI) improve, requiring fewer optimization steps. It would also be interesting to apply MILS to other modalities and tasks, such as for spatial and 3D tasks.

## Impact Statement

Our work builds on top of LLMs and other multimodal models, and as such, the strengths and weaknesses of such models would be reflected when used with MILS. This includes the biases present in such models. However, MILS being a training-free approach, does not learn any new biases, and can easily be upgraded with better and less-biased LLMs or multimodal models as they become available, without needing any re-training. While we do not foresee any negative societal impact directly enabled by MILS, any real-world applications should consider additional safety evaluations before deployment.

**Acknowledgements:** Authors would like to thank Xi Yin, Saketh Rambhatla, and the entire Meta AI team for many helpful discussions.

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

# A. Human evaluation Setup

We use Amazon Mechanical Turk (AMT) for human evaluation, with a setup similar to (Girdhar et al., 2024). As discussed in Section 4.4, we evaluate the enhanced image generation on two axes: quality and faithfulness to the text prompt. Screenshots of the evaluation interface are shown in Figure 13. The annotators are also given elaborate instructions with examples, as shown in Figures 14 and 15 for the faithfulness and quality evaluations respectively. We give each sample to three annotators and use majority vote to choose the winning model.

# B. LLM prompts

## B.1. Generating the initial candidate set

We bootstrap captioning tasks with an initial candidate set. We use the following strategies to create this initial candidate set.

**Image and video captioning:** We use the initial candidate set as generated in (Gandelsman et al., 2024). See Table 7 in their Appendix for the exact prompt. The overall idea is to take class labels from ImageNet (Deng et al., 2009) and prompt the LLM to generate 40 candidate captions based on the concept of the chosen class label. This process yields a set of around $30,000$ captions. This same set is used for the video captioning task.

**Audio captioning.** Since audio captions typically describe audible information, as opposed to visual information, we recreate the candidate set based on the above protocol with small modifications. We use class labels from AudioSet (Gemmeke et al., 2017) and generate 50 captions per audio label. We use Llama 3.1 and provide the following prompt:

> Generate 50 diverse descriptive captions for an audio clip that features the sound of {class_label}. Write a concise and vivid description of what can be heard in the clip, using complete sentences. For example:
> 1. A car drives by with its horn honking.
> 2. Children are playing and laughing in a park.
> 3. Heavy rain falls on pavement and roofs.
> 4. A crowd cheers and applauds at a sports event.
> Write the generation as if a person would write that after listening to the audio clip. Do not mention concepts that cannot be heard, like sunshine, star, any color or taste. Try to capture the main sounds and any background or accompanying noises in your caption, without referencing the fact that you're listening to an audio clip. Simply describe what can be heard. Put each description in a different line, with a counter at the beginning (e.g. '1. ...'), don't explain

> why, and don't combine two different concepts (with 'or' or 'and'), and keep it short 15-20 words.

where {class_label} is the AudioSet class name. The above prompt is similar to the prompt used in image and video captioning, adapted to mention the need for focus on audio. This process yields a candidate set of about $50,000$ captions.

## B.2. GENERATOR prompts

We use the following prompts in the GENERATOR. Here, {descriptions} denote the list of previous generations, along with the score for each of the generation from the SCORER. The scores are sorted, and we provide one score and text per row. In high-quality image generation, we also provide an additional {init_description} which is the original text prompt. {requested_number} is the number of new generations that we ask from the GENERATOR.

**Image Captioning:**

> You need to provide a short image description. I am providing to you a list of short image descriptions and scores. Higher score means that the image description characterizes the image better:
> {descriptions}
> Generate additional {requested_number} short image descriptions that you think that will maximize the score and fully capture the image. Be concrete and try to find elements that are unique to this image. You can introduce new elements to the descriptions, combine unique elements and objects from provided descriptions to form new descriptions, rephrase individual descriptions, drop elements, or simplify descriptions. Be creative and don't be afraid to come up with erroneous descriptions. Put each description in a different line, with a counter at the beginning (e.g. "1. ..."), and try to keep them very short (up to 10 words).

**Video captioning:**

> You need to provide a short video description. I am providing to you a list of short video descriptions and scores. Higher score means that the video description characterizes the video better:
> {descriptions}
> Generate additional {requested_number} short video descriptions that you think that will maximize the score and fully capture the video. Be concrete and try to find elements that are unique to this video. You can introduce new elements to the descriptions, combine unique elements and objects from provided

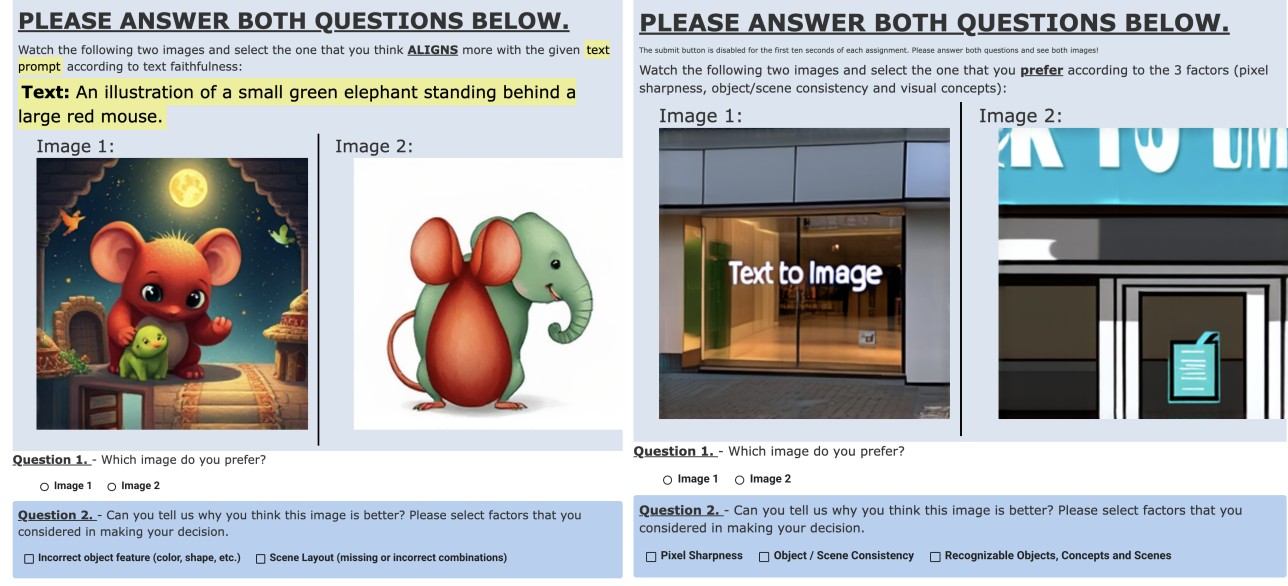

Figure 13: **Human evaluation question form for faithfulness (left) and quality (right).** We ask two questions. The first question asks which of the two images the rater prefers. The next question elaborates on the first response by asking the rater to list the reason for their preference, based on JUICE (Girdhar et al., 2024) evaluation protocol.

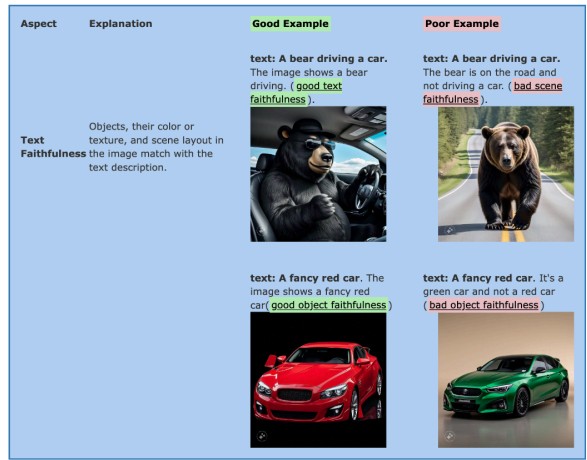

Figure 14: **High-quality image generation faithfulness examples.** We provide these examples to the annotators to judge the faithfulness. The generated image should have the correct object, color, texture or scene layout.

descriptions to form new descriptions, rephrase individual descriptions, drop elements, or simplify descriptions. Be creative and don't be afraid to come up with erroneous descriptions. Put each description in a different line, with a counter at the beginning (e.g."1. ..."), and try to keep them short up to 20-25 words.

**Audio captioning:**

> You need to provide a short audio description. I am providing to you a list of audio descriptions and scores. Higher score means that the audio description characterizes the audio better:
> {descriptions}
> Generate additional {requested_number} short audio descriptions that you think that will maximize the score and fully capture the audio. Be concrete and try to find elements that are unique to this audio. You can introduce new elements to the descriptions, combine unique elements and objects from provided descriptions to form new descriptions, rephrase individual descriptions, drop elements, or simplify descriptions. Be creative and don't be afraid to come up with erroneous descriptions. Put each description in a different line, with a counter at the beginning (e.g. "1. ..."), and try to keep them short (under 20 words).

**High-quality image generation:**

> You need to expand and rephrase the provided description for image generation to make the best image, by maximizing the image score: The description is: {init_description}
> Here are some example rephrases and the corresponding image scores:

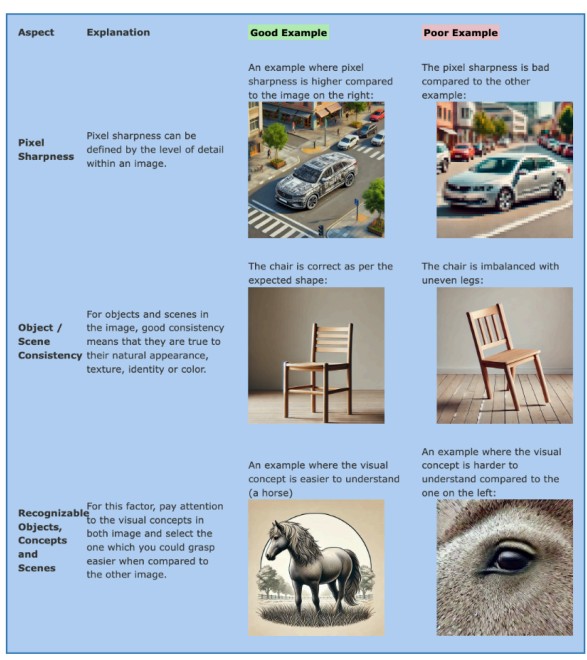

Figure 15: **High-quality image generation quality examples.** We provide these examples to the annotators to judge the quality. The resulting image should have high pixel sharpness, object and scene consistency, and the concepts should be recognizable.

| Model | Training data | SPICE |
|---|---|---|
| CLIP | WIT (Radford et al., 2021) | 8.5 |
| SigLIP | WebLI (Zhai et al., 2023) | **9.7** |
| MetaCLIP | MetaCLIP 400M (Xu et al., 2024) | 9.2 |
| DFN | DFN2B (Fang et al., 2024) | 9.3 |

Table 4: **Type of SCORER.** All the models are based on ViT-L/14 with different training data.

> {descriptions}
> Generate additional {requested_number} descriptions that will maximize the score. Be concrete and come up with different descriptions with various guesses for the possible way to rephrase and expand it, in a way that will maximize the score. You can introduce new elements to the descriptions, combine unique elements and phrasings from provided descriptions to form new ones, drop description parts, or simplify them. Be creative and don't be afraid to come up with erroneous descriptions. Put each instruction in a different line, with a counter at the beginning (e.g. "1. ..."), and keep them short (less than 77 words).

**Style transfer:**

> You need to generate instructions for image editing that minimize a pair of scores: I am providing you a list of example editing instructions and their pairs of scores.
> {descriptions}
> Generate additional {requested_number} editing instructions. Be concrete and come up with different instructions with various guesses for the possible edits that will minimize both of the scores. You can introduce new styles to the instructions, combine unique styles and textures from provided instructions to form new instructions, rephrase individual instructions, drop instruction parts, or simplify them. Be creative and don't be afraid to come up with erroneous editing instructions. Put each instruction in a different line, with a counter at the beginning (e.g. "1. ..."), and keep them short (less than 50 words).

**Cross-modal arithmetic:**

> I have an image description and an audio description that I want to combine together into a text description that will help an AI imagine that scene. As an example, if the caption says "Crane on a grass" and the audio says "An ocean with the waves crashing on shore" then you need to generate a text description like "Crane beside the shore with waves coming". The combinations can be imaginative and not necessarily true in real world. Here are the captions and the audio description:
> Image caption: {image_caption}
> Audio caption: {audio_caption}
> Generate the combined caption in a single sentence.

## C. Additional Results and Ablations

### C.1. Image Captioning

In Fig. 17, we show examples of image captioning on fantastical images from an image generation model. We see that the MILS model is able to generate captions that are more descriptive and accurate than the baseline model, even for images that are not realistic. This example also showcases the rigidity of memory-based models like MeaCap (Zeng et al., 2024).

### C.2. Video Captioning

Please see the attached supplementary video for video captioning results.

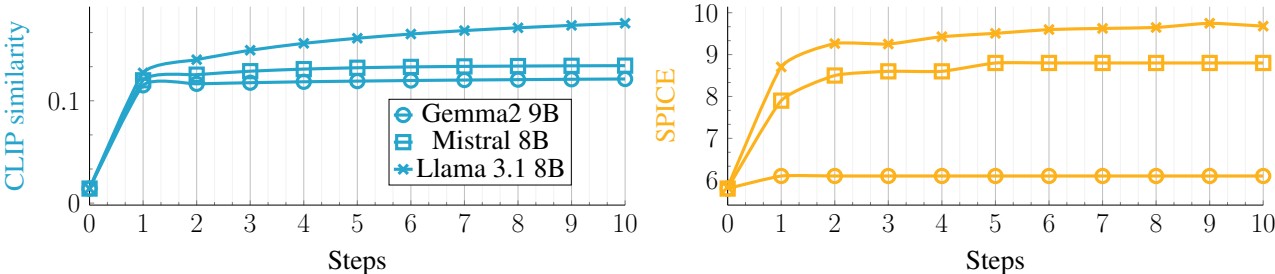

Figure 16: **Performance comparison with various LLMs as GENERATOR– Gemma2 9B, Mistral 8B, and Llama 3.1 8B, on CLIP similarity (left) and SPICE (right).** This trend shows that the performance improves over optimization steps, regardless of the choice of the LLM. In particular, Llama 3.1 8B gives the best performance, and is used primarily as the GENERATOR in our experiments.

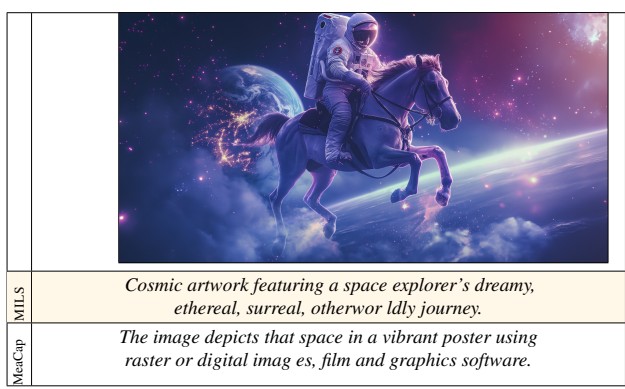

Figure 17: **Image Captioning using MILS**, compared to existing state-of-the-art zero-shot approach, MeaCap (Zeng et al., 2024) in captioning fantastical image. We see MeaCap is not able to generate the correct caption since it uses a memory that only contains captions for real-images.

### C.3. Audio Captioning

Please see the attached supplementary video for audio captioning results.

### C.4. Cross-Modal Arithmetic

Please see the attached supplementary video for cross-modal arithmetic results.

### C.5. Inference-time analysis

MILS' inference time is proportional to the number of iterations. As shown in Fig. 9, MILS performs well even with a few iterations, and the runtime will continue to decrease as more efficient and higher quality LLMs become available, that can reason in fewer steps. Moreover, the initial set size does not affect compute time much, as we can compute dot product similarity with the media features very fast on modern GPUs.

### C.6. Ablations

**Different types of GENERATOR and SCORER** We experiment with different open source LLMs, and CLIP models. We present our results in Figure 16 and Table 4.

First, we observe that while we see the same trend of performance improvement over steps for all LLMs we considered, the Llama model performed the best, with Mistral a close second. This shows that MILS generalizes to various different LLMs, and can expect to improve as LLMs advance further.

Second, we see in Table 4 that SigLIP is the strongest scorer for this task, when compared with models of the same parameter size. Again, the performance remains high for all scorers, showcasing the flexibility of MILS.

**Investigating convergence to the global maximum** As discussed in Section 3, we explore an $\epsilon$-greedy strategy to encourage exploration. This process includes low-score generations, to account for noise in the scoring process. However, we do not observe any gain using this idea. Regardless, MILS' overall use of a stable score is similar to other applications of optimization in machine learning.

