# OpenReview forum: "LLMs can see and hear without any training"
_ICML.cc/2025/Conference — ICML 2025 poster_

### Official Review · Reviewer_Deut · 2025-03-11

**Overall Recommendation:** 2

**Summary:**

The paper introduces Multimodal Iterative LLM Solver (MILS), a novel framework designed to provide multimodal capabilities, such as image, video, and audio captioning, without the need for task-specific training. The central innovation of MILS is its iterative optimization strategy, leveraging the inherent reasoning skills of LLMs. It operates by repeatedly using an LLM as a GENERATOR that proposes candidate solutions, which are then assessed by a SCORER, typically a pretrained multimodal model. The scored feedback informs subsequent generations in a loop until convergence. Key contributions include achieving emergent zero-shot generalization across different modalities, performing competitively with specialized models. MILS simplifies those multimodalities tasks through a straightforward approach..

**Claims And Evidence:**

Overall, the paper presents strong experimental evidence supporting many of its claims. However, some claims, particularly those regarding Cross-Modal Arithmetic and Generalization Ability, require further clarification and justification.

1. The paper claims that MILS allows arithmetic across modalities by converting them into text representations, combining them, and mapping them back. However, the authors do not clearly explain what these text representations are. Are they simply raw text formats? If so, what differentiates MILS from a pipeline consisting of a captioning model followed by a generation model? Without a clear distinction, the novelty and effectiveness of MILS in cross-modal arithmetic remain unclear.

2. The paper claims that MILS is conceptually simple and generalizable to many tasks. While the framework itself is modular, its success is heavily dependent on the choice of the GENERATOR and SCORER models. The effectiveness of MILS may vary significantly depending on the underlying models used, making it less generalizable than implied.

**Essential References Not Discussed:**

References are essential.

**Experimental Designs Or Analyses:**

1. The paper determines convergence by monitoring score similarity over iterations. However, this method assumes that a stable score directly correlates with optimal performance, which may not be valid.

2. It lacks established quantitative measures such as FID, or other diversity metrics, which would provide a more comprehensive analysis of generation quality and text-image alignment.

3. The effectiveness of MILS depends on the GENERATOR and SCORER models, but the paper does not provide enough ablation studies to analyze the contribution of different components. For example, it is unclear how different LLMs as GENERATOR impact the results.

**Methods And Evaluation Criteria:**

1. The paper uses score similarity to determine convergence, assuming that a stable score indicates an optimal solution. However, this approach may be insufficient and potentially misleading. A lack of score change does not necessarily imply the best possible result, it could indicate stagnation in a local minimum rather than true convergence.

2. The paper primarily evaluates image generation and image editing using PickScore and human preferences, but these alone do not fully assess generation quality. Metrics like FID could offer quantitative insights into visual realism and text alignment, while diversity metrics would help assess mode collapse.

**Other Comments Or Suggestions:**

The paper is well-written and organized, making it easy to follow the methodology and experimental setup.

**Other Strengths And Weaknesses:**

The paper presents a straightforward approach, but its novelty is limited. The proposed method essentially combines an LLM with a pre-trained multimodal model in an iterative optimization loop, which does not fundamentally differ from existing approaches that use LLMs and multimodal models together. While MILS formalizes this process as a simple framework, it lacks a clear distinction from standard prompting techniques or direct zero-shot inference using existing multi-models.

**Questions For Authors:**

The paper presents MILS as a zero-shot multimodal optimization framework, but it appears to function similarly to direct LLM prompting with a multimodal model as a filter. What distinguishes MILS from simply iterating over LLM-generated candidates and selecting the best one using a scoring model? A clearer articulation of its novelty is expected.

**Relation To Broader Scientific Literature:**

The paper extends test-time reasoning in LLMs by applying iterative refinement beyond text tasks. In zero-shot multimodal learning, prior captioning methods rely on gradient-based optimization, whereas MILS achieves zero-shot generalization purely at inference by combining LLMs with multimodal scoring models. The paper also contributes to multimodal representation learning, similar to ImageBind, by inverting multimodal embeddings into text, enabling cross-modal arithmetic and generation.

**Theoretical Claims:**

No theoretical claims in the paper.

---

> ### Author Rebuttal · Authors · 2025-04-01
>
> We thank the reviewer for your insightful review and helpful comments. We address all concerns below:
>
> ## Clarification of the text representation in cross modal arithmetic
>
> The representation is indeed the raw text. The novelty is in the inversion of the image into the caption itself. Using a captioning model would require training it on image-caption data. MILS can invert multimodal embedding spaces like CLIP into text *without needing any captioning training* and is completely gradient free, allowing for the inversion to discrete text as opposed to a continuous embedding space. Combining that with generation, it is able to perform cross-modal arithmetic, similar to ImageBind (Girdhar et al. 2023). However, by using text-space instead of the image embeddings in ImageBind, MILS is a lot more flexible–making it work with various image generation models, allows for user control etc (as we mention in L384 left).
>
> ## Generalizability of MILS
>
> While MILS’ performance does depend on the choice of LLM/scorer, it works well with a wide range of LLMs and scorers. Please see Fig. 16 and Tab 4 in appendix – while better scorer/generator give better results, MILS works reasonably well across all the combinations of LLMs (eg Llama, Mistral, Gemma) and scorers (CLIP, SigLIP, DFN), showing its generalizability.
>
> ## Convergence to the global maximum
>
> Great point! To avoid local maximas, we tried an experiment similar to noise injection in gradients from ML optimization.  As we mention in L134 (right column), we chose a portion of the candidates in an optimization step randomly, without considering their score. The idea is that, if a caption has a low score in the current round, a variant of it can still have a better score at a later round, effectively escaping the local maxima. However in practice, we did not notice a significant improvement.
>
> Regardless, MILS’ overall use of a stable score is similar to other applications of optimization in ML (e.g. neural network training) -- it could lead to a local optima but still be useful. As we show, MILS’ optimization leads to many useful applications. We will add more details and a discussion on this point in the appendix.
>
> ## Using FID as a metric
>
> In the image generation experiments, our goal is to generate images preferred by humans. Thus, human evaluation is the best fit for our setup.
>
> FID, on the other hand, has been shown to not be a reliable measure for human preference, as discussed in relevant papers ([Jayasumana et al. 2024 ‘Rethinking FIDs…’](https://arxiv.org/abs/2401.09603), [Parma et al. 2022, ‘On…GAN evaluation’](https://arxiv.org/abs/2104.11222), [Chong et al. 2020, ‘...unbiased FID…’](https://arxiv.org/pdf/1911.07023), [Borji et al. 2018, ‘Pros and Cons…’](https://arxiv.org/abs/1802.03446)). Large-scale image/video generation efforts (eg [SDXL](https://arxiv.org/abs/2307.01952), [Emu Video](https://arxiv.org/abs/2311.10709),  [MovieGen](https://arxiv.org/abs/2410.13720), etc) have found automatic metrics like FID do not correlate well with human judgment, and also primarily focus on human evaluation. SDXL mentions “we find these classical quantitative scores not to be suitable for evaluating the performance of foundational (text-to-image) DMs.” Consequently, we do not use FID as a metric in this work.
>
> ## It is unclear how different LLMs as GENERATOR impact the results….and other ablation studies
>
> We do provide experiments studying the effect of different kinds of  LLMs and Scorers in Appendix Fig. 16 and Tab. 4, respectively. We also study performance over different sized LLMs and Scorers in Fig. 12 (main paper). The performance is proportional to the relative strength of the LLM being used.
>
> Please let us know what other ablation studies are missing and we will clarify and add those.
>
> ## It lacks a clear distinction from standard prompting techniques or direct zero-shot inference using existing multi-models.
>
> We would love to get more details on specific models that the reviewer is comparing novelty to, and will be happy to compare to that.
>
> In general, multimodal vision-language models models are not *emergent zero-shot* (as our approach is), since they are trained for tasks like captioning. Hence, they generalize ‘zero-shot’ to a different data distribution. Our approach, on the other hand, generalizes to the completely new task of captioning itself – and captions images without ever being trained for it. We mention this in L071 left.
>
> ## How is it different from choosing the best generation using a scorer?
>
> This is exactly the first step of our optimization process, and the performance remains low without iterative refinement using the generator and the scorer (Fig. 9, 0th step).
>
> In short, our method consists of creating an initial set, followed by using the scorer to find top-K candidates, and iteratively feeding it back to the generator to create more candidates based on the score. This process is described in Sec. 3.

---

### Official Review · Reviewer_XRE1 · 2025-03-13

**Overall Recommendation:** 2

**Summary:**

This paper introduces MILS (Multimodal Iterative LLM Solver), which enables large language models (LLMs) to perform various multimodal tasks without any additional training. Through test-time optimization, it incrementally improves its outputs by leveraging two key modules (GENERATOR and the SCORER), by generating a set of outputs and feedback with scores to iteratively refine the outputs. Experimental studies show that MILS can handle image, video, and audio captioning as well as generative tasks such as text-to-image creation and style transfer, all without requiring task-specific pretraining.

**Claims And Evidence:**

The paper claims that MILS (Multimodal Iterative LLM Solver) demonstrates emergent zero-shot multimodal capabilities through an iterative optimization framework leveraging pre-trained LLMs and CLIP-based multimodal scorers. However, I feel it is inadequately supported by the presented evidence. To convincingly establish this emergent reasoning capability, the authors must explicitly analyze and experimentally distinguish the incremental performance benefits provided specifically by iterative reasoning via LLM-generated candidates compared to existing methods that already effectively leverage pre-trained multimodal embedding representations.

- First, the paper does not clearly distinguish how much iterative reasoning via the LLM contributes to the observed zero-shot performance beyond what has already been demonstrated by existing memory-based methods such as MeaCap. Recent works like MeaCap have already shown strong zero-shot multimodal capabilities by directly leveraging pre-trained multimodal embeddings (e.g., CLIP, SigLIP, ImageBind). Given this prior work, it is essential for MILS to provide explicit evidence demonstrating that iterative candidate generation through LLM reasoning meaningfully enhances performance beyond what pre-trained multimodal embeddings alone achieve. However, the authors do not provide a clear comparison of the additional benefits specifically attributable to iterative optimization via LLM-generated candidates.

- Additionally, in the presented experimental results (e.g., Table 1), MILS does not achieve clear superiority over memory-based approaches such as MeaCap, particularly regarding critical evaluation metrics like CIDEr and BLEU. For instance, MILS attains a CIDEr score of 33.3, which is substantially lower than MeaCap’s 42.5 (and 51.7 when using high-quality SS1M memory). Higher scores (42.1 and 50.6) of DeCap  (which is only cited) are also missing. While the paper selectively emphasizes metrics such as METEOR and SPICE, even in these semantic metrics, MILS either achieves performance similar to or only marginally better than MeaCap.

**Essential References Not Discussed:**

The paper explicitly cites DeCap (ICLR, 2023) in its references but omits DeCap from main experimental comparisons (such as Table 1).
- Reported numbers of DeCap correspond to Table 1 are 42.1 and 50.6 CIDEr, which are significantly higher than 33.3 of MILS. Including DeCap in experimental comparisons is essential for clearly validating the effectiveness of MILS’s iterative candidate optimization.

**Experimental Designs Or Analyses:**

I have several concerns about the experimental design and analyses.

1. Inadequate Comparison with MeaCap
- The paper’s experimental comparisons with MeaCap are insufficient. MeaCap employs a memory retrieval-based approach, which offers fundamentally different scaling properties and cost structures compared to the iterative methodology of MILS. While MeaCap’s performance can significantly improve with larger and higher-quality memories (as evidenced by the high CIDEr score using SS1M), MILS is constrained by the inherent limitations of LLM’s candidate generation. Even increasing the number of iterations or candidates does not seem to surpass these fundamental limitations. This disparity between the two approaches is not adequately addressed or discussed in the experimental design.

2. Lack of Analysis on Error Accumulation
- The authors do not provide an analysis of error accumulation—a critical issue in iterative methods. With each iteration, there exists a risk that erroneous candidates may be generated and, through the feedback process, inadvertently reinforced. Such errors can be caused by either Generator or Scorer, and can be worse during iteration adversarial to CLIP scores.

3. Missing Computational Efficiency Evaluation
- Given that MILS necessitates repeated calls to an LLM, I have concerns regarding inference time and computational cost. However, the paper does not present any rigorous evaluation or discussion on computational efficiency. Due to the fundamental limitation of LLM capability, I suspect MILS has to scale up LLM to achieve fine-quality outputs. I also found the video provided in the supplementary material provided by Llama 405B model.
- When we consider MILS requires around 10 (or more) steps for optimization, it seems too expensive compared to the existing memory-style training-free approaches.

**Methods And Evaluation Criteria:**

The methodological approach of MILS is based on iterative optimization. It leverages a process in which a large language model (LLM) generates multiple candidates and a SCORER provides repeated evaluations to address multimodal tasks. However, the iterative optimization employed by MILS is fundamentally restricted by the candidate generation capacity of the LLM.
- During each iteration, MILS generates tens to hundreds of candidates. This process implies that the ultimate quality of the captions is inherently capped by the LLM’s linguistic generation capabilities. In contrast, the memory-based approach adopted by MeaCap capitalizes on a vast external corpus that stores extensive world knowledge, which can be retrieved and filtered during caption generation.
- Consequently, an increase in both the size and quality of the memory can directly and substantially enhance caption performance. For example, according to MeaCap, when employing a large-scale, high-quality memory such as SS1M, the CIDEr score can rise to 42.5. By comparison, MILS, even when limited to around 10 iterations, converges at a CIDEr score of 33.3, demonstrating significantly lower performance relative to SS1M.
- This discrepancy shows a fundamental drawback: the iterative candidate approach in MILS appears to have intrinsic performance ceilings that cannot be overcome merely by increasing the number of iterations or candidates. However, the authors do not sufficiently discuss or analyze this limitation.

**Other Comments Or Suggestions:**

see the above sections.

**Other Strengths And Weaknesses:**

Strengths
- This paper clearly reveals the performance capabilities and limitations of pre-trained LLM-based training-free methods across various multimodal tasks. By evaluating emergent zero-shot capabilities in multimodal tasks such as video captioning and image generation, MILS provides strong baseline performance achievable purely through iterative optimization of LLM-generated candidates and multimodal embeddings.

Weakness
- MILS inherently requires extremely large-scale pre-trained LLMs to achieve competitive performance. As explicitly shown in the supplemental experiments on video captioning tasks, MILS utilizes exceptionally large-scale LLMs such as LLaMA 400B. Such huge model sizes impose considerable computational costs and resource constraints, severely limiting practical feasibility and scalability. Specifically, the iterative optimization approach of MILS inherently restricts meaningful performance gains to the expensive strategy of continually scaling up the size of the underlying LLM. In contrast, memory-based approaches (e.g., MeaCap) inherently offer a much more scalable and cost-effective means of improving performance by simply increasing the size and quality of external memory. Given these considerations, while MILS can indeed serve as a useful baseline reference for multimodal tasks beyond standard captioning benchmarks, it appears clear that, in practice, memory-based methods would offer a far more effective solution due to their superior scalability and cost-efficiency.

**Questions For Authors:**

see the above sections.

**Relation To Broader Scientific Literature:**

The key contributions of this paper align closely with recent works exploring training-free and zero-shot multimodal approaches such as MeaCap and DeDap, specifically those relying on pre-trained large multimodal embedding models and LLMs.
- For example, although MILS does not employ external memory directly, MeaCap similarly avoids task-specific training and relies heavily on pre-trained multimodal embedding models (e.g., CLIP) for evaluation. The key difference lies in MILS’s iterative candidate generation approach, where it repeatedly generates and scores multiple candidate captions, refining them iteratively. In contrast, MeaCap utilizes a single-step retrieval-and-generation pipeline, relying solely on high-quality external memory to enhance caption quality. So, I think MILS's K-candidates play a similar role in MeaCap's memory. It can be useful when we even do not have any external texts to build such memory, but  I think high-quality memory can perform much better than the proposed method.

**Theoretical Claims:**

N/A

---

> ### Author Rebuttal · Authors · 2025-04-01
>
> We thank the reviewer for their detailed feedback. We address all concerns below. In particular, we highlight the benefits of our iterative approach vs MeaCap, which is designed specifically for image captioning. We also clarify a possible misunderstanding regarding the use of large 405B LLM–except the qualitative examples shown for cross-modal arithmetic, all other results/examples in the main paper+supplementary use Llama 8B.
>
> ## Benefits compared to MeaCap/DeCap
>
> a) **Iterative optimization generalizes to more tasks**
>
> MeaCap/DeCap are specific to image captioning. They leverage a corpus of image captions from real-world datasets (e.g. CC3M). In contrast, MILS’ initial set does not contain captions from any dataset. Consider the task of captioning fantastical images from an image generation model. Retrieval based methods like MeaCap/DeCap would struggle since the CC3M captions will be very out-of-distribution and need significant modifications. To illustrate this, we ran both on [this image](https://www.freepik.com/free-ai-image/spaceman-riding-horse-outer-space_222195825.htm), and got the following outputs:
> ```
> MeaCap: The image depicts that space in a vibrant poster using raster or digital images, film and graphics software.
> MILS: Cosmic artwork featuring a space explorer's dreamy, ethereal, surreal, otherworldly journey.
> ```
> Clearly, MILS captures the essence of the image much better than MeaCap. We will add more such comparisons to the final paper.
>
> b) **MILS avoids hand-crafted design and embraces end-to-end optimization**
>
>
> MeaCap relies on a hand-designed subject-predicate-object based refinement, followed by iterative CBART sentence generation. MILS does not enforce any such constraints, and end-to-end optimizes the LLM generations, directly using feedback from the pre-trained multimodal embedding model. In this sense it can be thought of as a simplification and generalization of MeaCap, making it applicable to five more tasks than just image captioning.
>
>
> c) **Experimental comparison with MeaCap and Decap (Tab. 1)**
>
> We report comparison with *training-free versions* of all methods: 42.1 and 50.6 CIDEr in DeCap require training the LM on captions from image/video-captioning datasets. In contrast, MILS is completely training-free (TF). Moreover, MILS’ 33.3 CIDEr, outperforms MeaCap results that we could reproduce from the published code, which is 26.0 for MeaCap-TF (vs 42.0 reported in the paper)–we confirmed with the authors this variance was expected for the TF version.
>
>
> ## “Quality of the captions is inherently capped by the LLM’s linguistic generation capabilities”
>
> LLMs have very strong linguistic generation capabilities, as evident from their ability to generate stories, rhymes, etc; even conversing with humans such that many consider it to [have passed the turing test](https://www.nature.com/articles/d41586-023-02361-7). The external corpus in MeaCap–CC3M–contains 3M captions, which one could argue is much smaller than the linguistic ability and world knowledge stored in the LLM, if measured by their training data or parameters.
>
> In fact, using CC3M gives MeaCap an advantage over MILS since it leverages captioning-specific knowledge, while MILS relies only on a general-purpose LLM.
>
> ## MeaCap improves performance by increasing memory, “MILS appears to have intrinsic performance ceilings”
>
> MILS scales well too! We obtain clear improvements using LLMs/Scorers that are bigger (Fig 12), better (Appendix Fig 16, Tab 4), or using a larger initial set (Fig 10), with no ceilings observed thus far.
>
>
> ## Error accumulation analysis
>
> MILS performs iterative refinement, i.e. optimization, as opposed to iterative generation—the error will reduce over iterations as shown in our results over optimization steps (Fig. 9 and Fig. 8, 11). Errors would accumulate if we were fixing some part of our output at a given iteration and generating the remainder based on that—we are globally improving the output until it converges.
>
> ##  Computational efficiency/“seems too expensive with 10 steps for optimization”
>
> The inference time is proportional to the number of iterations. The model performs well even with few iterations–please refer to the performance curves in Fig. 9, which shows even in 3 steps the performance is within 90% of the best. The iteration knob further allows MILS to trade-off quality vs speed, for time-sensitive applications. We expect the speed to improve and number of steps required to reduce as LLMs become more efficient and powerful.
>
> ## “Use of extremely large pretrained LLMs such as Llama 405B”
>
> This is a possible misunderstanding. We only use 8B LLM models for all experiments except multi-modal arithmetic (Fig. 7). All captioning tasks use 8B LLM. In fact, we only used 405B for multi-modal arithmetic since we could easily use API calls for the qualitative results; using the 8B variant also yields similar results. We will update the paper with 8B for arithmetic for consistency.

---

> > ### Comment · Reviewer_XRE1 · 2025-04-03
> >
> > Thank you for your rebuttal. However, it does not fully address my concerns, particularly regarding the empirical evidence and specific comparisons requested earlier; the scalability and efficiency of the proposed method in comparison to existing memory-based approaches such as MeaCap.
> >
> > For example, the current CIDEr score of MILS (33.3) is substantially lower than MeaCap's reported scores (42.5, and up to 51.7 with SS1M), highlighting a clear performance gap. While memory-based methods have the advantage of easily scaling by increasing memory quality, MILS heavily relies on scaling up the size of the LLM to enhance performance, raising concerns about computational efficiency and scalability.
> >
> > Additional quantitative evidence or analyses demonstrating how effectively MILS can scale in practice to match or surpass memory-based approaches within reasonable computational constraints would be beneficial to readers.

---

> > > ### Author Response · Authors · 2025-04-03
> > >
> > > Dear reviewer XRE1,
> > >
> > > Thank you so much for your response
> > > 1) We would like to reiterate, MeaCap obtains 26.0 CIDEr when we run inference with MeaCap’s released code, which is lower than MILS’ 33.3 CIDEr. We confirmed with the MeaCap authors on a **public forum** that such variance is expected. We are unable to share a link to that forum discussion to preserve anonymity, but will add it to the final paper. We recap the performance of MILS compared to MeaCap in the following table.
> > >
> > >    | Method                         | BLEU₄ | CIDEr | METEOR | SPICE |
> > >    |-------------------------------|:-----:|:-----:|:------:|:-----:|
> > >    | MeaCap*ᴛꜰ (Zeng et al., 2024) (reproduced) |  4.5  | 26.0  | 14.1   |  9.4  |
> > >    | **MILS**                      | **8.0** | **33.3** | **15.0** | **9.6** |
> > >
> > >    Additionally, please consider the qualitative results of MILS compared to MeaCap, as shown above in the rebuttal for a fantastical image, and in the Paper (Fig. 3). Automatic metrics like CIDEr only provide limited signal when evaluating generative models (as also observed with FID and FVD for image and video generation, please see our response to Reviewer Deut for a discussion). Qualitatively, we observed a clear improvement with MILS compared to MeaCap and other prior work.
> > >
> > >
> > > 2) On scaling, again reiterating from the rebuttal, MILS is highly scalable. As shown in the paper Fig. 10, MILS’ performance improves significantly with the initial set size. The initial set can be thought of as similar to MeaCap’s memory, hence showing a similar scaling behavior as in MeaCap. In addition, MILS shows promising scaling performance with multiple other axes, such as LLM size (Fig. 12), scorer size (Fig. 12), LLM type (Fig. 16), and scorer type (Tab. 4), arguably making it a lot more scalable than MeaCap.
> > >
> > > Please let us know if either of these aspects are unclear, and we’d be happy to clarify further.
> > >
> > > Thank you again,
> > >
> > > Authors

---

### Official Review · Reviewer_qksR · 2025-03-14

**Overall Recommendation:** 4

**Summary:**

The authors propose MILS: Multimodal Iterative LLM Solver, a simple method that claims to use the reasoning abilities of textual LLMs to have impressive zero-shot performance on multimodal tasks. At a high level, this involves a Generator model, which is either a text-LLM or a text-LLM chained to another system such as an image generation model and a Scorer, which evaluates the quality of the generations. By optimizing the generator to produce outputs that are preferred by the scorer, it is found that the method is competive against purpose-built models for image, video and audio captioning, and also improves generative capabilities for text-to-image generation. The authors also discuss cross-modal arithmetic where they are able to combine different modalities using MILS

**Claims And Evidence:**

The authors claim that their method is extremely generalizable and only requires a generator and scorer, which can be basically any system that generates output for a task and one that can evaluate them. This is well-justified by examples of image/audio/video captioningn and generative tasks such as image generation and style transfer.

**Essential References Not Discussed:**

I did not find any major references missing

**Experimental Designs Or Analyses:**

The experiments are set up in an appropriate manner, with detailed explanation of bootstrapping, set up and output generation provided across a variety of tasks. The experiments are mainly just examples of the method working well, rather than trying to question and/or investigate the method itself, though.

**Methods And Evaluation Criteria:**

The authors compare to appropriate models using standard metrics on standard benchmark datasets.

**Other Comments Or Suggestions:**

None

**Other Strengths And Weaknesses:**

Strengths:
1. The method is simple, plug-and-play and effective, showing improved performance of the method on a variety of tasks.
2. The method also generalizes across different textual LLMs, and scales to their intelligence/reasoning abilities.

Weaknesses:
1. There is little, if any, deep investigation into the reasoning capabilities of the LLM that cause this to work. I would like to see a more detailed deep dive into the optimization process and it's potential brittleness when coming up against harder tasks.
2. I am not sure if this is necessarily novel and therefore appropriate for ICML. The novelty is very much at the system-level and the idea of a scorer giving feedback to a generator, while applied well, is not new to the ML literature in general.

**Questions For Authors:**

I do not have any major questions for the authors. This is a well-written paper and shows a practical method that enables us to make existing systems better without expensive training runs.

**Relation To Broader Scientific Literature:**

The fundamentally plu-gan-play nature of this method implies that it is both deeply embedded in the general generative deep learning literature while also standing somewhat apart from it by being a gradient-free method that is plug and play.

**Theoretical Claims:**

The paper does not make any theoretical claims of note

---

> ### Author Rebuttal · Authors · 2025-04-01
>
> We thank the reviewer for the insightful review and their willingness to accept our paper. We are glad they found our method generalizable, claims well justified, and improved performance on a variety of different tasks. We address all concerns next:
>
> ## Investigation into the reasoning capabilities of the LLM that cause this to work
> This is a great suggestion and we’ll attempt this for the final paper. Understanding the reasoning capabilities of LLMs is indeed a very fast evolving field of research, and many of the discoveries there would also help better understand MILS. For instance, deeper insights into MILS can be obtained by taking into account the reasoning traces available in recent LLMs such as DeepSeek-R1. Another interesting exploration could be in the LLM-scorer interface, for instance experimenting with the sensitivity of the LLM to the resolution of the scorer outputs. In general, LLMs as optimizers are relatively easier to probe compared to more opaque optimizers, given their ability to explain decisions in plain language. We believe this ability will help MILS expose new avenues for research both in LLM reasoning and multimodal understanding.
>
>
> ## Novelty statement compared to other ML literature
>
> Ours is the first work to leverage test-time reasoning capabilities of LLMs to solve multimodal tasks, obtaining state-of-the-art results across a wide range of computer vision/multimodal benchmarks: image, video and audio captioning, enhanced image generation (outperforming very strong models like FLUX), style transfer and cross-modal arithmetic. As reviewer 934y also notes on the novelty, this approach enables pure text-only LLMs to see and hear without needing any training on that kind of data. Our novelty also lies in the simplicity of our framework that does not need any task-specific modifications. We do take inspiration from standard ML literature where the concept of scoring functions and optimization is used extensively, however to our knowledge, it hasn’t been used in this way for test-time optimization, using LLMs, to solve multimodal tasks.
>
>
> If there is specific prior work the reviewer would like us to compare to, please let us know and we would be happy to provide more comparisons to that.

---

> > ### Comment · Reviewer_qksR · 2025-04-02
> >
> > Thank you for your comment, I appreciate the explanation, and will keep my score as 4/Accept.

---

### Official Review · Reviewer_934y · 2025-03-16

**Overall Recommendation:** 3

**Summary:**

This paper presents an iterative solver to enable pure language LLMs to "perceive" visual or audio signals through trial and error using discriminative scorers. Specifically, the authors construct a feedback loop between generators (LLMs) and discriminative ranker (e.g., SigLIP for image captions): first, the LLM first output initial random guess lists by prompting; next, the score ranks guess lists with regard to images/audio, and feed guess lists with scores back into the LLM; then, the LLM analyze the guess lists with scores and iteratively refine new guess lists into the "right" direction. The authors also generalize this framework to audio caption, image generation and style transfer tasks, and explore other cross-modal tasks with the help of ImageBind.

**Claims And Evidence:**

The claims, LLMs can see and hear without any training, can be supported by clear and convincing empirical evidence presented in this paper. Indeed, LLMs can see and hear by correctly prompting based on feedbacks in a loop style. We can say this paper presents a gradient-free optimization to the score by prompting candidates.

**Essential References Not Discussed:**

I do not find any specific references missing in this submission. But it would make this paper more sound to discuss more gradient-free optimization.

**Experimental Designs Or Analyses:**

Most experimental designs and analyses are sound and valid in my opinion. A minor concern is the absence of inference time analysis concerning the number of iterations and the initial set size, which is crucial for real-world applications.

**Methods And Evaluation Criteria:**

This method present a novel approach to make pure LLMs to see and hear without any training on image/audio-language pair samples. That said, I have concerns whether proposed MILS have real application potentials compared to other multimodal LLMs. While loop methods between generators and scores being under-explored before, MILS performance seems relatively lagging behind common pipelines to multimodal understanding. I understand that authors' goal is not achieve new state-of-the-art performance, but the looping mechanism between generators and scores seem time-costing and highly inefficient. Indeed, I find MILS do have potentials in generating high-quality image generation/style transfer samples in Sec 4.4 and 4.5 to serve as post-training samples for further fine-tuning nowadays pipelines. Now back to my question, for multimodal understanding, what is the strength or potential of MILS rather than proposing yet another way to do it?

**Other Comments Or Suggestions:**

NA

**Other Strengths And Weaknesses:**

Overall I think this paper presents a "yet-another-approach" to leverage pure LLM to understanding multimodal signals without any training/fine-tuning on paired samples. This paper is indeed novel but with some minor concerns in real application potentials (See Methods And Evaluation Criteria).

**Questions For Authors:**

Q.1. As mentioned in Experimental Designs Or Analyse, the authors might provide inference time analysis and its comparison with standard multimodal understanding pipelines.

Q.2. I feel MLIS quite limited in "caption" tasks. Actually, it seems that the capability of MLIS as a gradient-free optimization is "bounded" by the scorer capability. In the paper case, the scorer, SigLIP, provide caption and visual similarity. My concern is, whether MLIS can solve other multimodal questions beyond captioning, like spatial referring? No experiments needed but I'd like to see authors' insights into it.

**Relation To Broader Scientific Literature:**

In my view, this paper presents a gradient-free optimization for scores by the scorer by prompting candidate sets.

**Theoretical Claims:**

There are no theoretical claims nor proofs in this submission.

---

> ### Author Rebuttal · Authors · 2025-04-01
>
> We thank the reviewer for their insightful review and willingness to accept the paper. We are glad you found our work novel, and claims supported by clear and convincing empirical evidence. We address all the remaining concerns below:
>
>
> ## For multimodal understanding, what is the strength or potential of MILS rather than proposing yet another way to do it?
>
> Great point! We agree -- one can train specific multimodal models for tasks that have enough training data that are hyper-optimized for that specific task. For online production use cases that require millisecond-level optimization, training a specific model for that task might indeed be the way to go.
>
> MILS, on the other hand, can supercharge research exploration or applications that may not require such a high level of runtime optimization. For example, MILS can quickly provide a strong multimodal model for a new task, without any training, completely zero-shot. This could be useful to researchers as an initial baseline for any new task they might want to solve. It could also serve as an easy and effective approach to interpret new models a researcher develops. Consider the use case of interpreting the strengths/weaknesses of a new multimodal embedding model and comparing it to CLIP/SigLIP etc. MILS can enable that interpretability by helping invert each model’s embedding space into text (without any training that might bias this inversion). The produced image captions for different embedding models could be very helpful for a researcher to better understand the behaviors of these models. Finally, MILS can be used as an offline data generator, which can then be used to train or distill optimized multimodal models. Consider for instance the enhanced image generation task – MILS can be used to generate large amounts of <text, improved image> pairs, which can be used to tune a text-to-image model for improved quality.
>
> ## Q1. Inference time analysis
>
> MILS’ inference time is proportional to the number of iterations. As shown in Fig. 9, MILS performs well even with a few iterations (eg 3), and the runtime will continue to decrease as more efficient and higher quality LLMs become available, that can reason in fewer steps. Moreover, the initial set size does not affect compute time much, as we can compute dot product similarity with the media features very fast on modern GPUs. We will add additional runtime analysis and comparisons with similarly sized task-specific multimodal understanding models in the final paper.
>
>
> ## Q2. Can MILS solve tasks like spatial referring?
>
> Certainly! We can generate box proposals and referring expressions (e.g. a class label or a short description) with the LLM, compute box-level CLIP similarity score by computing CLIP features on the cropped box, and use that as the score to give feedback to LLM. The LLM will then produce new box proposals and referring expressions. We can then repeat this iterative process as proposed in MILS. The scoring can further be made more efficient by pooling part of the full image’s spatial CLIP features that correspond to the box, and matching that to text.
>
> Overall, as long as we design a scorer for a task, which can evaluate the correctness of the LLM generation with respect to the input (can be any modality), we can integrate that task into MILS. It can be similarly extended to segmentation, object detection in 3D, etc.
>
> ## Discuss more gradient-free optimization papers
>
>
> That is a great suggestion! We will add more references and discussions to gradient-free optimization works to the final paper.

---

> > ### Comment · Reviewer_934y · 2025-04-09
> >
> > Thank authors' rebuttal and appreciate the insights. I have no further questions and still feel positive about this paper. Therefore I'd keep my original rating as it is.

---

### Decision · Program_Chairs · 2025-05-01

**Decision:**

Accept (poster)

**Comment:**

This paper proposes a training-free approach to extend LLM for multi-modal tasks such as image captioning, text-to-image generation, etc. The basic idea is to use LLM to generate prompts that are fed to to an additional multi-modal scorer. The paper receives mixed scores. In general, most reviewers consider the problem and proposed approach to be novel, the presentation is clear, the gradient free approach to extend LLM with multi-modal ability is interesting, and the experiments are extensive. At the same time, reviewers also raised multiple concerns. For example, inference time complexity seems to be high, the underlying working mechanism of the proposed method is not very clear, the reliance of the performance on specific generator and scores, as well as the convergence issue of the proposed optimization method. While most of the concerns have been addressed in the rebuttal, some reviewers remain unconvinced on several aspects. Particularly, some reviewer believe one critical issue of the proposed method is the scalability. At the current stage, it is not clear whether the method can be scaled to outperform existing baselines such as MeaCap. I share a similar concern with this, and also have a little question on the reasonability of the proposed method since it seems to have limited application scenarios in real problems.

Overall, I consider this paper to be borderline. It is acceptable depending on the room in the program.